# Novel Thermosensitive and Mucoadhesive Nasal Hydrogel Containing 5-MeO-DMT Optimized Using Box-Behnken Experimental Design

**DOI:** 10.3390/polym16152148

**Published:** 2024-07-29

**Authors:** Pablo Miranda, Analía Castro, Paola Díaz, Lucía Minini, Florencia Ferraro, Erika Paulsen, Ricardo Faccio, Helena Pardo

**Affiliations:** 1Unidad de Nanotecnología, Instituto Polo Tecnológico de Pando, Facultad de Química, Universidad de la República, Montevideo 91000, Uruguay; pmiranda@fq.edu.uy (P.M.); acastro@fq.edu.uy (A.C.); 2Biomind Labs, Brookfield Place, 181 Bay Street, Suite 1800, Toronto, ON M5J 2T9, Canada; paola.diaz@nordelis.com (P.D.); lucia.minini@nordelis.com (L.M.); florencia.ferraro@nordelis.com (F.F.); 3Instituto de Ingeniería Química, Facultad de Ingeniería, Universidad de la República, Montevideo 11800, Uruguay; erikap@fing.edu.uy; 4Área Física, DETEMA, Facultad de Química, Universidad de la República, Montevideo 11300, Uruguay; rfaccio@fq.edu.uy

**Keywords:** Poloxamer 188, Poloxamer 407, HPMC, tryptamines, intranasal formulation, sustained release

## Abstract

We present the development and characterization of a nasal drug delivery system comprised of a thermosensitive mucoadhesive hydrogel based on a mixture of the polymers Poloxamer 407, Poloxamer 188 and Hydroxypropyl-methylcellulose, and the psychedelic drug 5-methoxy-N,-N-dimethyltryptamine. The development relied on a 3 × 3 Box-Behnken experimental design, focusing on optimizing gelification temperature, viscosity and mucoadhesion. The primary objective of this work was to tailor the formulation for efficient nasal drug delivery. This would increase contact time between the hydrogel and the mucosa while preserving normal ciliary functioning. Following optimization, the final formulation underwent characterization through an examination of the in vitro drug release profile via dialysis under sink conditions. Additionally, homogeneity of its composition was assessed using Raman Confocal Spectroscopy. The results demonstrate complete mixing of drug and polymers within the hydrogel matrix. Furthermore, the formulation exhibits sustained release profile, with 73.76% of the drug being delivered after 5 h in vitro. This will enable future studies to assess the possibility of using this formulation to treat certain mental disorders. We have successfully developed a promising thermosensitive and mucoadhesive hydrogel with a gelling temperature of around 32 °C, a viscosity close to 100 mPas and a mucoadhesion of nearly 4.20 N·m.

## 1. Introduction

In the last years, studies on psychedelics tryptamines like N,-N-dimethyltryptamine (DMT) or 5-methoxy-N,-N-dimethyltryptamine (5-MeO-DMT), have become of great interest to medical research due to their promising properties for treating neurodegenerative and psychiatric disorders [1]. 5-MeO-DMT is a natural compound found in a wide variety of plant species and is the major active ingredient in Bufo alvarius toad venom [2,3,4]. It is also synthesized by the human retinal and pineal gland in trace amounts [5]. Historically used in recreational and spiritual settings, this molecule has awakened therapeutic interest in the treatment of psychiatric conditions. Recent studies have demonstrated the ability of psychedelics to treat brain injuries and a variety of psychiatric disorders, such as depression, anxiety, headache, Alzheimer disease and obsessive-compulsive disorder [6,7,8,9,10,11,12].

In terms of pharmacological effects, 5-MeO-DMT is known to be a potent psychedelic substance that induces an altered state of consciousness primarily through a pathway involving activation of a serotonin 5-HT1A receptor. The O-demethylation of 5-MeO-DMT is well known to be catalyzed by the cytochrome P450 2D6 (CYP2D6) enzyme to produce bufotenine, an active metabolite that proved to be a potent ligand of 5-HT2A receptors [13]. In addition, it undergoes a rapid first-pass metabolism effect due to deamination mediated by the enzyme monoamine oxidase A (MAO-A), which inactivates this molecule [14]. Concomitant use of a MAO inhibitor is essential to avoid inactivation of 5-MeO-DMT after oral administration. However, such a combination can result in synergistic or antagonistic responses, causing serious or even fatal toxicity [15].

Intranasal administration of drugs is an effective alternative to achieve therapeutic concentrations directly into the brain through the olfactory and trigeminal nerves, bypassing the blood-brain barrier and reducing the hepatic first-pass effect [16]. This ensures greater bioavailability and at the same time faster onset of action. Administration through this route is simple, allowing dose reduction and minimizing adverse effects compared to systemic and oral drug delivery approaches [17]. However, the nasal route has some disadvantages, such as limited retention time and rapid elimination through mucociliary clearance [18].

For nasal administration, the use of a defined-dose spray is beneficial, though other factors must be considered during formulation. For instance, the formulation should be isotonic with nasal fluid and have reduced viscosity to enhance drug retention on the mucosa. Intranasal administration of a pure 5-MeO-DMT solution is generally avoided due to these limitations and the anatomical connection between the nasal cavity and pharynx, which further reduces retention time. To address these issues, we propose formulating 5-MeO-DMT as a thermosensitive and mucoadhesive gel instead of a simple solution, thereby enhancing both retention time and bioavailability within the nasal cavity, optimizing its therapeutic potential.

Scientific literature indicates that the half-life of 5-MeO-DMT and tryptamines, in general, is only a few minutes [19,20]. Current research on the potential use of tryptamines for treating mental disorders focuses on maintaining therapeutic concentrations for extended periods. This prolonged delivery method reduces the number of required administrations compared to conventional release formulations. By minimizing dosing frequency, this approach aims to improve patient compliance and ensure a more consistent therapeutic effect, which is particularly important for mental health treatments.

Thermosensitive and mucoadhesive hydrogels are being extensively investigated as alternative formulations for the delivery of a wide range of drugs, including antimicrobials, analgesics, and anticancer agents [21,22,23]. Hydrogels are polymeric matrices that can retain a large amount of water or biological fluids under physiological conditions. Their unique sol-gel transition properties in response to specific stimuli, such as temperature, pH, ionic strength, light, electromagnetic radiation, and biomolecules, make them excellent candidates for drug delivery systems. Temperature-sensitive hydrogels represent one of the most widely used types of environmentally responsive polymer systems since their transition from sol to gel mediated by a temperature increase provides an attractive approach to in situ gel formation.

This behavior is possible due to the unique properties of the constituent polymers, particularly their amphiphilic nature which influences the solubility and gelling capability of the hydrogel. The sol-gel transition is mediated by changes in intermolecular interactions due to temperature variations. At lower temperatures, hydrophilic interactions dominate, allowing the polymer to dissolve in water and remain in a liquid state. Additionally, repulsive forces between hydrophilic segments keep the polymer chains separated. As the temperature increases, these repulsive forces diminish, allowing hydrophobic forces to become more prominent, causing the polymer chains to associate and form a gel [24].

Poloxamers are triblock copolymers of poly(ethylene oxide)-poly(propylene oxide)-poly(ethylene oxide) (PEO-PPO-PEO) with special thermosensitive gelling properties, low toxicity, and high water solubility. Poloxamer 407 (P407) and Poloxamer 188 (P188) are among the most popular synthetic polymers displaying such characteristics [25]. Hydrogels prepared using poloxamers tend to increase the water solubility of otherwise insoluble drugs due to their surfactant properties [26]. These polymers are widely utilized in medicine for their distinct properties and functionalities. Poloxamer 407 acts as a viscosant agent and exhibits thermosensitive properties, forming gels at body temperature, which is advantageous for sustained or controlled drug delivery systems. It also favors drug solubility and stability in formulations [27,28,29]. Poloxamer 188 provides solubility enhancement of poorly soluble drugs and is also employed in cell cryopreservation to protect cells during freezing and thawing, with potential applications in tissue repair [28,29].

A formulation designed as a multidose nasal spray with mucoadhesive properties that also presents the phenomenon of gelling in situ activated by temperature changes can be beneficial to accomplish both a good retention time and a correct drug delivery on the nasal mucosa. This behavior can be obtained by including polymers such as polyacrylates, cellulose derivatives (hydroxypropyl-methyl cellulose (HPMC), hydroxypropyl cellulose, methyl cellulose, carboxymethyl cellulose), and natural polymers (chitosan, alginate) [30,31,32,33,34,35]. Under normal cold storage conditions (or even at room temperature), the polymeric mixture (thermosensitive polymer plus mucoadhesive polymer) disperses in the formulation, forming a colloidal solution. This allows dosing in the form of a spray [36,37]. The small droplets generated during multidose spraying can adhere to the nasal mucosa thanks to the mucoadhesive and thermosensitive properties.

In the present study, we aimed to develop, characterize and optimize a thermosensitive Poloxamer-based nasal hydrogel for the delivery of 5-MeO DMT. We carried out the optimization using a three-level three-factorial Box-Behnken experimental design to evaluate the effects of the amount of P407, P188 and HPMC in the viscosity, gelation temperature and mucoadhesion of the different formulations prepared.

## 2. Materials and Methods

### 2.1. Chemicals

5-MeO-DMT (99% purity) was supplied by Biosynth (Compton, UK). Poloxamer P407 and Poloxamer P188 were gifted by BASF (Ludwigshafen, Germany). Hydroxypropyl Methylcellulose (HPMC) K100 was acquired from Colorcon (Buenos Aires, Argentina). Porcine stomach mucin type II was purchased from Sigma–Aldrich (St. Louis, MO, USA). All other reagents were of analytical grade.

### 2.2. Preparation of Thermosensitive Gels

First, 0.45 g of glycerin were dissolved in 25 mL of saline solution (0.90% *w/v* sodium chloride). Then 50 μL of benzalkonium chloride (10% *v*/*v*) were added. After that, the solution was cooled down in an ice water bath, to a temperature of 8 °C. Polymers and the drug were then slowly added under vigorous stirring, in a set order. First, the HPMC K-100 was added. After total dissolution of the polymer, 0.50 g of 5-MeO-DMT were added. Finally, P188, and P407 were incorporated. This final mixture was stirred for 2 h, and then stored at 8 °C overnight to get a crystal clear colloidal solution.

### 2.3. Experimental Design

A 3X3 Box-Behnken response surface methodology with 3 center replicates was used to optimize the hydrogel formulation. Constraints were applied to maximize mucoadhesion, while keeping viscosity lower than 200 mPas and Tgel below 35 °C. Box-Behnken design consists of a set of replicated center points, and points taken at the middle of each edge of the multidimensional cube defining the selected workspace. As a result, a response surface is therefore created, which can be represented by the following nonlinear quadratic model: Y = b_0_ + b_1_X_1_ + b_2_X_2_ + b_3_X_3_ + b_12_X_1_X_2_ + b_23_X_2_X_3_ + b_13_X_1_X_3_ + b_11_X_2_ + b_22_X_2_ + b_33_X_2_. In this case, the variable Y is the response arising from the combined contributions of each factor at the different levels tested, b_0_ is the model intercept and b_1_–b_33_ correspond to the adjusted regression coefficients for each individual factor. Data processing was carried out using Design Expert 11 software (StatEase, Minneapolis, MN, USA). Table 1 resumes the different formulations prepared, as well as the factors evaluated and their corresponding levels.

### 2.4. Gelation Temperature Determination

For this study, we used a custom ‘‘Visual Inversion Method’’. Approximately 5 mL of each formulation were transferred to test tubes placed in a Maxturdy 30 shaking water bath (Daihan Scientific, Daedeok-gu, Korea). Then, samples were heated up in increments of 0.50 °C from 20 to 40 °C, with an equilibration period of 5 min. A digital thermometer was inserted in a blank saline solution to track temperature inside the tubes. The study finished for each formulation, when a complete stop in flow was observed after rotating the test tube 90°.

### 2.5. Rheological Studies

Dynamic viscosity determination of all formulations was carried out using a Visco QC 100 viscometer (Anton Paar, Graz, Austria) equipped with a B-CC18 spindle. The data acquisition interval was 1 min. Viscosity measurements were performed immediately after withdrawing the samples from the fridge, at a temperature of approximately 8 °C.

### 2.6. Mucoadhesion Analysis

The study was performed with a TA-XT plus texture analyzer (Stable Micro Systems, Godalming GU7 1YL, UK) set to adhesiveness mode. Mucin type II discs attached horizontally to the lower end of a cylindrical probe using double sided adhesive tape. Mucin discs (10 mm in diameter) were prepared by compression of porcine stomach mucin Type II (100 mg) using a ring press. 100 μL of each sample were placed in a metallic platform underneath the probe, which was then forced down at a defined rate of 2 mm/s until contact with the sample was achieved. Probe and sample then remained in contact for 180 s applying a constant downward force of 0.50 N at a speed of 0.10 mm/s, with a trigger force set to 0.01 N. Then, the probe was raised at a rate of 0.50 mm/s to a predetermined distance of 5 mm. Mucoadhesion work (measured in N·m) was determined as the area under the curve of the resultant force–distance curve, using the Exponent Connect software, version number 6.2.

### 2.7. Confocal Raman Spectroscopy

Measurements were performed with a WITec Alpha 300-RA Raman-Confocal Microscope (WITec GmbH, Ulm, Germany). The main standalone componentes (P407, P188, HPMC, 5-MeO-DMT) and the optimized hydrogel formulation were analyzed. All samples were positioned on the microscope stage, and Raman spectra were acquired with an excitation laser wavelength of 532 nm. Spectra were obtained as an average of a set of 100 measurements, each with an integration time of 0.50 s.

### 2.8. In Vitro Drug Release Study

Drug release was evaluated on three replicates of the optimized hydrogel formulation. First, 100 mg of gel (containing approximately 1600 μg of 5-MeO-DMT) were placed in 2 kDa pore size Slide-A-Lyzer G3 Dialysis Cassettes (ThermoScientific, Boston, MA, USA). The cassettes were then introduced in a beaker containing 400 mL of a 0.90% *w/v* sodium chloride solution, heated up to 35 °C in an ES-20 Orbital Shaker Incubator (Biosan, Riga, Latvia). Finally, 1 mL samples were drawn with reposition at set time intervals to determine drug concentration. Drug determination was performed on a Dionex Ultimate 3000 HPLC instrument equipped with a diode array detector (Thermofisher Scientific, Boston, MA, USA). The selected HPLC column was a 150 × 4.60 mm Zorbax 300 Extend-C18 with a particle size of 3.50 μm (Agilent, Santa Clara, CA, USA). The mobile phase consisted of a 70/30 water/acetonitrile mixture containing 0.10% of trifluoroacetic acid. The injection volume was 20 μL (full loop mode), temperature 40 °C, the flow rate 0.75 mL/min and the total runtime of the analysis was 5 min. UV-Vis spectra in the range 200 nm to 450 nm were recorded. Detection was carried out at 279 nm wavelength. Acquisition and processing were performed with Chromeleon 7.2.10 Chromatography Data System (Thermofisher Scientific, Boston, MA, USA).

## 3. Results and Discussion

### 3.1. Experimental Design

The independent variables (factors) selected for the study were the amounts of P407, P188 and HMPC respectively (expressed in *w/v* %). In order to achieve an optimal formulation, an experimental design was carried out using Design Expert 11 and applying a 3 × 3 Box-Behnken model with 3 responses. The 15 factor combinations (12 factorial points plus 3 center points) tested are shown in Table 2, alongside their respective results.

The optimization constraints were defined considering the gel was intended for mucosal drug delivery. For Tgel, we considered that normal nasal temperature is on average 35 °C. For viscosity, we needed the gel to be fluid, in order to allow a correct spraying and also to avoid disrupting mucosal ciliary movement. Ideally, viscosity should not exceed the 200 mPas [36,37]. On the other hand, the idea of a mucoadhesive gel is to increase the contact time between the mucosa and the active ingredient. For that reason, we set the constraint for mucoadhesion to maximize, and we allowed the software to explore the design space considering Tgel and viscosity under 35 °C and 200 mPas respectively.

#### 3.1.1. Model Fitting

In order to determine suitable mathematical expressions for all responses, we first fitted full quadratic models using Design Expert 11 and then conducted a manual backward elimination process. In each step, the least significant term was removed until every single remaining term was statistically significant (*p* < 0.05). Estimation of significance was conducted using analysis of variance (ANOVA). Model fitting was assessed via adjusted r^2^, predicted r^2^ and Lack of Fit test respectively. Table 3 summarizes these results.

After completing this process, the best fit for both Tgel and mucoadhesion was a quadratic model. For viscosity, the best fit turned out to be a two factor interaction model. In all cases, the predicted r^2^ fits were very close to the adjusted r^2^. Table 4 summarizes the results, and Figure 1 depict a correlation graph between predicted and actual data.

The equations are in agreement with the behavior expected from scientific literature. For example, HPMC is supposed to bestow mucoadhesive properties, but is not reported to be a thermosensitive polymer [33]. The fitted model for Tgel does not show HPMC to be a significant component of its behavior, but it correctly includes both poloxamers. In this case, P407 is apparently the most statistically significant on gelation temperature. On the other hand, HPMC should play an important role in viscosity and also in mucoadhesion. Both models correctly include this polymer in their corresponding equation. For example, in the case of viscosity, HPMC and P407 have the most statistically significant contribution. For mucoadhesion, it is the combination of these two polymers that is most statistically significant, despite being all 3 polymers included in the equation.

#### 3.1.2. Response Surface for Tgel

The sensitivity of the response variable to changes in the input factors, can be analyzed with the perturbations plot for each model. This provides insights into how variations in the input factors impact the response, also allowing the identification of interactions. Figure 2 depict the perturbation plots for Tgel (A), viscosity (B) and mucoadhesion (C).

In the case of Tgel, we can see that an increase in the amount of P407 reduces in a nonlinear way the temperature at which the colloidal solution turns into a gel. On the other hand, an increase in P188 has the opposite effect on Tgel, but with a linear correlation and to a much lesser degree than P407. We can also see there is an interaction between both factors, since the corresponding curves intersect each other. For viscosity, higher quantities of the 3 polymers seem to have a directly proportional effect on this variable. It is the amount of P407 followed by that of HPMC, the factors impacting viscosity the most. We can also see there is an interaction between all factors, since the corresponding curves intersect each other. Finally, for mucoadhesion, higher amounts of HPMC have a positive effect on this property as expected. P407 on the other hand, has a negative linear impact (inversely proportional), since increasing amounts of this polymer tend to decrease mucoadhesion, and also at higher degree compared to the positive effect of HPMC. It is also worth mentioning that P188 has a nonlinear negative effect on mucoadhesion. It seems to be that an equilibrium concentration window exists, after which increasing or decreasing the amount of P188 starts reducing mucoadhesion significantly. We can also see there is an interaction between both factors, since the corresponding curves intersect each other. Optimization was carried out following the constraints explained before. In brief, we wanted to maximize mucoadhesion, while keeping Tgel below 35 °C and viscosity under 200 mPas. The rationale behind this is the fact that the most important point to cover is the residence time, for which mucoadhesion has to be the highest possible. Then, we also needed the formulation to become a hydrogel very quickly at body temperature. The lower the temperature, the faster the solution turns into a hydrogel. However, we also needed to maintain a balance with viscosity. Lowering Tgel too much (adding more P407) would cause a severe increase in viscosity, something that is not desirable for this development. We know from the perturbation analysis that mucoadhesion is positively correlated with increases in HPMC and decreases in P407. On the other hand, since this last polymer has the highest impact on viscosity, reducing its amount in the formulation would also contribute to increasing the gel fluidity. This hypothesis was correctly supported by the response surface graphs and contour plot (all of them shown considering the reference factor at the corresponding medium level) for Tgel, viscosity and mucoadhesion, which are depicted in Figure 3, Figure 4 and Figure 5 respectively.

Depending on the system to be optimized and the constraints set in place, the optimization procedure arising from the response surface methodology can show different results. Sometimes there will be several possible combinations that satisfy the specified optimization criteria. This will all depend on the extent of the workspace fitting such criteria determined by the software. In some cases, many of the solutions provided will be very close to one another, not being significantly different in terms of the final experimental result or from a practical point of view. Sometimes the predicted optimal formulation could be a set of factor combinations not explored in the workspace. In such cases, preparation and testing of such formulation is required to validate the predictive power of the model. However, there will also be cases where the optimal formulation could even be one of the points already studied as part of the experimental design. In our case, the optimization model predicted that the best formulation that could fulfill the preselected requirements would be that containing 15% P407, 9.10% P188 and 1% HPMC *w/v* %, with a 0.966 desirability according to the software. This formulation is strikingly close to point 11 of the experimental design (15% P407, 10% P188 and 1% HPMC). In total, the software displayed 49 different solutions with desirabilities ranging from 0.408 to 0.966 respectively. Point 11 from the experimental design appears as an approximation in solution number 19 (15% P407, 9.97% P188 and 1% HPMC), with a desirability of 0.962. If we look at the predicted results displayed by the model, we can see there is no significant difference for any of the individual responses (Tgel, viscosity and mucoadhesion) if we consider the 0.966 top result and point 11 from the experimental design (which has a calculated desirability of 0.962). This is depicted in Table 5.

### 3.2. Raman Spectroscopy

To assess the intermolecular interactions involved in the hydrogel formation, Raman spectra were acquired from both individual components and their combinations. In Figure 6, Raman spectra of P407 (a), P188 (b), HPMC (c), 5-MeO-DMT (d), Glycerol (e), and the optimized hydrogel are presented. The distinctive Raman peaks of poloxamers are attributed to C-O-C bending at 848 and 1145 cm^−1^. Also, C-H wagging and rocking vibrational modes at 1284 cm^−1^, with CH bending and stretching modes clearly observed at 1483 and 2885 cm^−1^ [38,39]. For HPMC, bands corresponding to symmetric C–O–C appear at around 1122 cm^−1^ and also at 1370 cm^−1^ associated with COH bending, as well as at 1458 cm^−1^ due to the CH_2_ scissor vibrational mode [40].

For 5-MeO-DMT, Raman spectrum exhibits the most characteristic peaks associated with the indole ring in the 1361, 1447, and 1548 cm^−1^ regions [41,42]. Additionally, the stretching vibrational modes of the CH_2_ and CH_3_ of the methoxy group are situated in the range of 2828–3056 cm^−1^ [43]. Differences in peaks previously defined for HPMC, poloxamers and 5-MeO-DMT are discernible in the Raman spectra of the hydrogel (as can be seen in Figure 6). These outcomes corroborate the formation of the hydrogel, facilitated by hydrophobic and/or hydrophilic interactions, such as low energy Van der Waals forces or hydrogen bonds.

On the hydrogel Raman spectrum, a broad multimode OH stretching band can also be observed between 3100 and 3600 cm^−1^, this band corresponds to the free water present in the suprastructure of the hydrogel (see in Figure 6, peaks located at 3247 cm^−1^ and 3419 cm^−1^) [44].

Raman confocal spectroscopy also offers the possibility to visually determine which parts of the samples correspond to the compounds being analyzed. In Figure 7, several squares are presented. The single-color ones (c, d, e, f, g) represent each of the hydrogel’s main components analyzed. On the other hand, the first one (a) is the sample section analyzed, and (b) is a mosaic of all the signal contributions in the hydrogel. In other words, it shows the overall homogeneity of the sample. We can clearly see that all the compounds are correctly mixed, confirming that the gel is homogeneous.

### 3.3. In Vitro Drug Release Study

We conducted this experiment assuming a solubility limit of 32.70 µg/mL at pH 7.4 for 5-MeO-DMT based on online data available in Pubchem (Aqueous Solubility from MLSMR Stock Solutions), presented by the Burnham Center for Chemical Genomics. We also defined Sink conditions in such a way that the concentration of the study would never rise higher than 20% of the theoretical solubility limit [45]. This ensures that no saturation effects are present, so that the release profile obtained reflects only the effect of the formulation. In this case, the 20% limit for Sink conditions would be 6.74 µg/mL. The optimized hydrogel contains 0.50 g of 5-MeO-DMT in a total mass of 32.20 g. That is 15.53 mg of 5-MeO-DMT/g of hydrogel. Therefore, 100 mg of hydrogel contains approximately 1.55 mg of the drug. If we divide this by 400 mL (total volume of this drug release study), we get that the maximum achievable theoretical concentration is around 3.88 µg/mL. The release profile obtained was linear, as expected when this kind of experiments are conducted in sink conditions. As depicted in Figure 8, after 5 h, the amount of 5-MeO-DMT released was on average (considering 3 replicates) 73.76%, demonstrating that the formulation can be used to achieve a sustained release of the drug.

## 4. Conclusions

The primary objective of this study was to develop, characterize, and optimize a thermosensitive Poloxamer-based nasal hydrogel for the delivery of 5-MeO DMT. Formulation optimization utilized a three-level, three-factorial Box-Behnken experimental design. The results indicated that P407 significantly influenced both gelation temperature and viscosity, while HPMC also played a crucial role in viscosity control. Synergistic effects of P407 and HPMC were observed in enhancing mucoadhesion properties, as revealed by the experimental design outcomes. The optimized hydrogel formulation consisted of 15% *w/v* P407, 9.10% *w/v* P188, and 1% *w/v* HPMC.

Raman spectroscopy provided detailed insights into the molecular interactions contributing to hydrogel formation. Characteristic bands associated with hydrophilic and hydrophobic bonds confirmed significant chemical interactions between the polymers and the drug. Moreover, the spectroscopic analysis demonstrated the uniformity of component distribution within the hydrogel, crucial for ensuring consistent drug release.

In terms of drug release profile, a linear correlation was observed under sink conditions, with approximately 74% of the drug released within 5 h. These findings suggest that the developed hydrogel holds promise for the controlled intranasal delivery of 5-MeO-DMT.

## 5. Patents

A patent application related to this work has been filed with the United States Patent and Trademark Office. Date of filing: 30 December 2021. https://patents.google.com/patent/EP4159192A1/en.

## Figures and Tables

**Figure 1 polymers-16-02148-f001:**
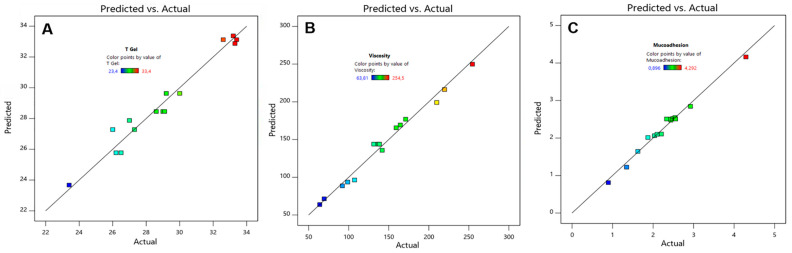
Predicted versus actual plot obtained by the Box-Behnken Design for hydrogel. gelation temperature (**A**), viscosity (**B**) and mucoadhesion (**C**). The color code represents predicted value points in the lower (bluish), medium (greenish), and high (reddish) parts of the continuous results interval.

**Figure 2 polymers-16-02148-f002:**
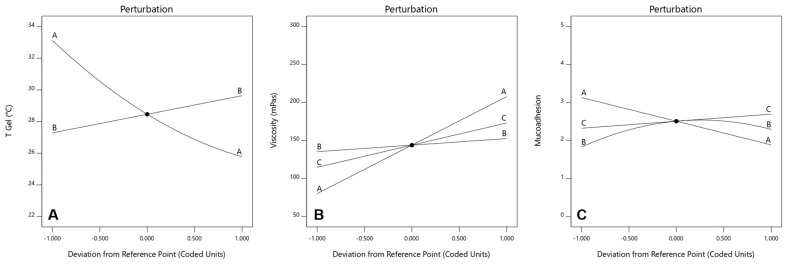
Perturbation plots for Tgel (**A**), viscosity (**B**) and mucoadhesion (**C**). The different curves show the sensitivity of the optimization responses (Tgel, viscosity, mucoadhesion) to changes in concentration levels of P407 (**A**), P188 (**B**) and HPMC (**C**) respectively.

**Figure 3 polymers-16-02148-f003:**
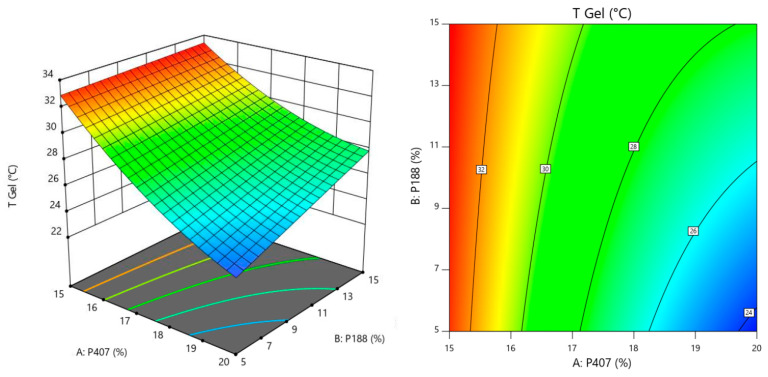
3D response surface and contour plots generated from the Box-Behnken design to examine the effects of P188 and P407 concentrations on Tgel.

**Figure 4 polymers-16-02148-f004:**
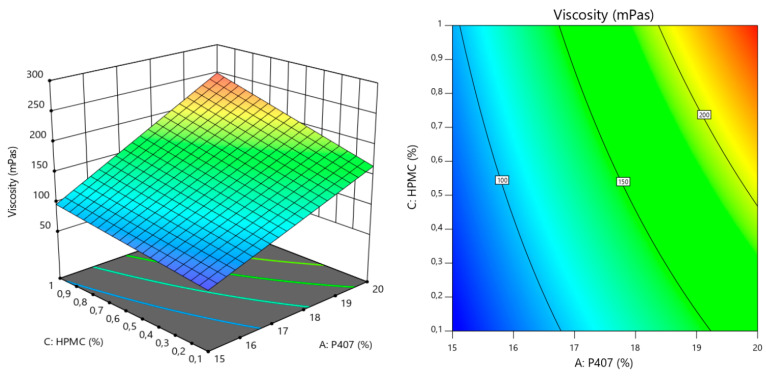
3D response surface and contour plots generated from the Box-Behnken design to examine the effects of HPMC and P407 concentrations on viscosity.

**Figure 5 polymers-16-02148-f005:**
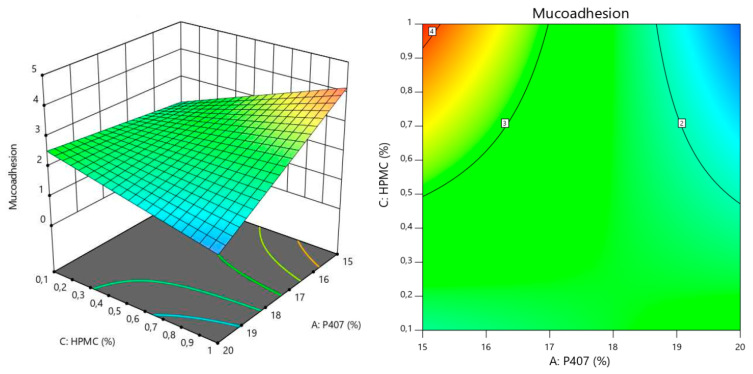
3D response surface and contour plots generated from the Box-Behnken design to examine the effects of HPMC and P407 concentrations on mucoadhesion.

**Figure 6 polymers-16-02148-f006:**
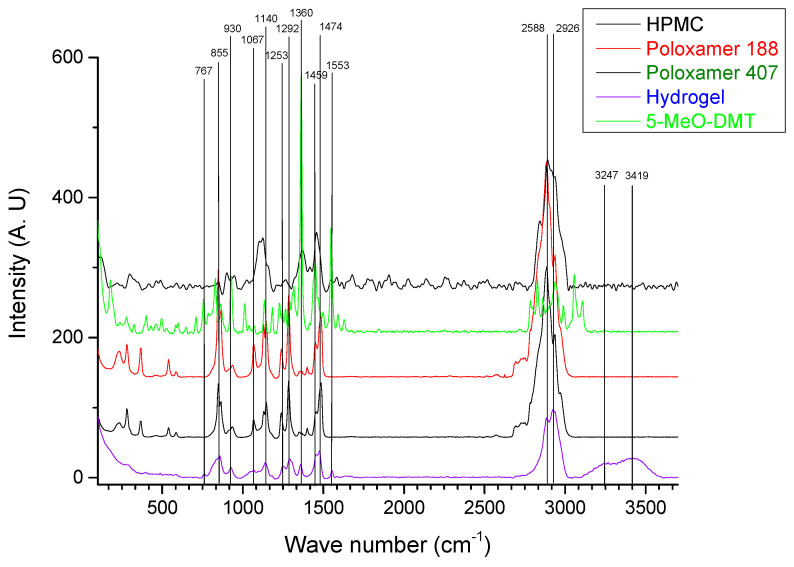
Raman spectra of individual and combined hydrogel components.

**Figure 7 polymers-16-02148-f007:**
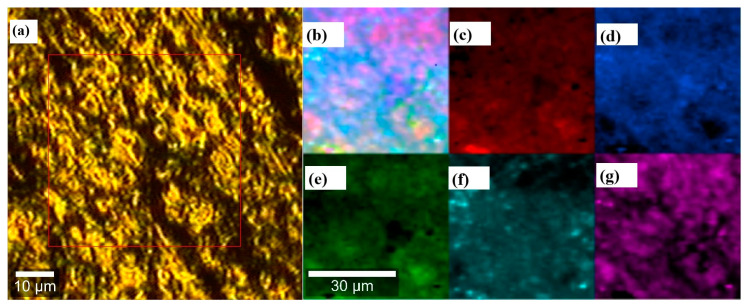
Raman confocal microscopy of sample section analyzed (**a**), hydrogel (**b**), HPMC (**c**), P188 (**d**), P407 (**e**), 5-MeO-DMT (**f**), water free (**g**).

**Figure 8 polymers-16-02148-f008:**
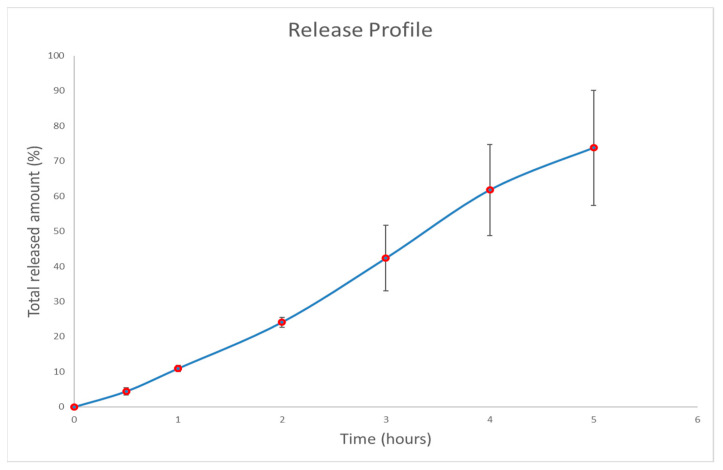
Drug release profile of 5-MeO-DMT from hydrogel at pH 7.4 and 35 °C.

**Table 1 polymers-16-02148-t001:** Independent and dependent variables in Box-1 Behnken design.

	**Levels**		
**Independent variables**	**Low**	**Medium**	**High**
X_1_ = P407 (% *w*/*v*)	15	17.50	20
X_2_ = P188 (% *w*/*v*)	5	5	15
X_3_ = HPMC (% *w*/*v*)	0.10	0.55	1

**Dependent variables**	**Constraints**		
Y_1_ = Tgel (°C)	Y_1_ ≤ 35		
Y_2_ = Viscosity (mPas)	Y_2_ ≤ 200		
Y_3_ = Mucoahesion (N·m)	Maximize		

**Table 2 polymers-16-02148-t002:** Box-Behnken experimental design and observed responses for the hydrogel containing 5-MeO-DMT.

Experiment	P407 (%)	P188 (%)	HPMC (%)	Tgel (°C)	Viscosity (mPas)	Mucoadhesion (N·m)
1	20	10	1	26.2	254.5	1.347
2	20	5	0.55	23.4	209.9	0.896
3	17.5	10	0.55	28.6	131.3	2.556
4	17.5	5	1	26	171.1	1.873
5	17.5	5	0.10	27.3	98.7	1.624
6	15	15	0.55	33.2	92.1	2.489
7	17.5	15	1	29.2	164.4	2.439
8	20	10	0.10	26.5	159.3	2.541
9	15	10	0.10	33.4	63.81	2.10
10	15	5	0.55	33.3	69.42	2.923
11	15	10	1	32.6	107.3	4.292
12	20	15	0.55	27	219.6	2.035
13	17.5	15	0.10	30	141.8	2.204
14	17.5	10	0.55	29	136.9	2.384
15	17.5	10	0.55	29.1	138.3	2.335

**Table 3 polymers-16-02148-t003:** Model fitting parameters.

**Tgel**	***p* Value**
Model	<0.0001
A (P407)	<0.0001
B (P188)	0.0007
AB	0.0235
A^2^	0.0199
Lack of Fit Test	0.1116

**Viscosity**	***p* value**
Model	<0.0001
A (P407)	<0.0001
B (P188)	0.021
C (HPMC)	<0.0001
AC	0.0158
BC	0.0188
Lack of Fit Test	0.1337

**Mucoadhesion**	***p* value**
Model	<0.0001
A (P407)	<0.0001
B (P188)	0.0008
C (HPMC)	0.0031
AB	0.0002
AC	<0.0001
B^2^	0.0001
Lack of Fit Test	0.5154

**Table 4 polymers-16-02148-t004:** Summary of the regression analysis of the three responses for the quadratic models.

	Fit Equation
**Tgel (°C)**	Y_1_ = 113.43 − 7.77X_1_ − 1.06X_2_ + 0.07X_1_X_2_ + 0.16X_1_^2^ Adjusted r^2^ = 0.9487 Predicted r^2^ = 0.8929
**Viscosity (mPas)**	Y_2_ = −275.67 + 19.21X_1_ + 4.76X_2_ − 80.85X_3_ + 11.49X_1_X_3_ − 5.53X_2_X_3_ Adjusted r^2^ = 0.9747 Predicted r^2^ = 0.9480
**Mucoadhesion (N·m)**	Y_3_ = −2.51 + 0.23X_2_ + 0.18X_3_ + 0.39X_1_X_2_ − 0.85X_1_X_3_ − 0.45X_2_^2^ Adjusted r^2^ = 0.9731 Predicted r^2^ = 0.9387

**Table 5 polymers-16-02148-t005:** Comparison of predicted solutions from the optimization model with Formulation Number 11.

Formulation	Tgel (°C)	Viscosity (mPas)	Mucoadhesion (N·m)
Best solution (0.966 desirability)	32.6798	97.0418	4.17737
Solution number 19 (0.962 desirability)	32.7233	96.3711	4.16372
Point 11 (0.962 desirability)	32.7249	96.3465	4.16271

## Data Availability

Data will be made available on request.

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
