# Peer review of "Novel Thermosensitive and Mucoadhesive Nasal Hydrogel Containing 5-MeO-DMT Optimized Using Box-Behnken Experimental Design"

_polymers, 2024, doi:10.3390/polym16152148_

Round 1

Reviewer 1 Report

Comments and Suggestions for Authors

The development of prolonged-release systems for medical applications is of great relevance. The gel composition created by the authors appears promising and interesting. However, to enhance accessibility and comprehensibility, the authors should consider incorporating a few key suggestions.

It is recommended that the introduction be rewritten and that the following questions be answered in the text:

  1. Prolonged release is clearly justified, for example in wound care, to maintain a constant concentration of the substance locally, but what is the point of prolonged delivery of a psychedelic drug to the brain? 

  2. Why is the pure 5-MeO-DMT solution not currently injected intranasally?

  3. Can the developed system be considered as a universal system and be used for the delivery of other drugs?

  4. Using an developed gel with a psychedelic drug raises a number of medical questions: In the event of an overdose or unpredictable reaction by the patient, can the gel be quickly removed? How to control the gel's distribution in the sinuses? How to assess how much of the substance has entered the body and how much has leaked out of the nose to the outside?

  5. Are there other intranasal prolonged release systems in medicine?

  6. How are the selected polymers (P407, P188, HPMC) currently used in medicine? Are there any examples of the use of a blend of these polymers?

  7. Are there any studies of the effects of these polymers on 5-MeO-DMT?

In reading the main body of the article, the following questions and comments arose:

  1. What is "PE"? (line 10) This abbreviation occurs only once and is not deciphered

  2. Paragraph experiment design (3.1). The authors use mass-to-volume ratios (w/v %) as the main characteristic for polymer composition selection. This characteristic is well suited for dissolution of substances in pure water. Perhaps for a multicomponent system it is better to use mass fraction (w/w %). This approach may be important because Section 2.2 shows that the developed system contains glycerin, sodium chloride, benzalkonium chloride, 5-MeO-DMT and water. As the concentration of the main polymers increases, the mass fraction of the other components of the system changes. It turns out that the other components are variables and they are left out of the authors' view. For example, viscosity is affected by the mass fraction of water, while Tgel can be affected by the mass fraction of salt. This issue can be avoided if the total mass of P407, P188, HPMC remains constant in all experiments. The authors should give arguments why the ratio of the other components of the system is not important.

  3. How were the starting concentrations of polymers at center points chosen? (P407=17.5; P188=10; HPMC=0.55)

  4. Figure 2 may become clearer if you sign which curve refers to which polymer

  5. On lines 327-328, the authors write "In Figure 6, Raman spectra of P407 (a), P188 (b), HPMC (c), 5-MeO-DMT (d), Glycerol (e), and the optimised hydrogel are presented", but in Figure 6 there are no letters a-e and no Glycerol is present

  6. Line 340 has a reference to Figure 10, but there is no such figure present

  7. In 3.4 the authors study the release of the substance from the gel. It is written that 73.76% was released, but it is not clear whether this is a enough or not for medical use. The section will be richer if the authors write how much 5-MeO-DMT is usually injected into a patient and how much gel is needed for this amount of substance.

Author Response

Reviewer #1

The development of prolonged-release systems for medical applications is of great relevance. The gel composition created by the authors appears promising and interesting. However, to enhance accessibility and comprehensibility, the authors should consider incorporating a few key suggestions.

It is recommended that the introduction be rewritten and that the following questions be answered in the text:

Comments 1. Prolonged release is clearly justified, for example in wound care, to maintain a constant concentration of the substance locally, but what is the point of prolonged delivery of a psychedelic drug to the brain? 

Response 1. Thank you for your insightful question. According to the scientific literature, the half-life of 5-MeO-DMT and tryptamines, in general, is only a few minutes (1, 2). Current research on the potential use of tryptamines for treating mental disorders is focused on maintaining therapeutic concentrations for extended periods. This prolonged delivery method reduces the number of administrations required compared to conventional release formulations. By minimizing the frequency of doses, this approach aims to improve patient compliance and ensure a more consistent therapeutic effect, which is particularly important for mental health treatments.

Comments 2. Why is the pure 5-MeO-DMT solution not currently injected intranasally?

Response 2. Thank you for your question. Intranasal administration of pure 5-MeO-DMT solution is not currently performed due to several limitations. One primary disadvantage is the rapid elimination of the compound caused by ciliary movement and continuous mucus production. Additionally, the anatomical connection between the nasal cavity and the pharynx further reduces retention time.

To address these issues, we proposed formulating 5-MeO-DMT as a thermosensitive and mucoadhesive gel instead of a simple solution. This advanced formulation enhances both retention time and bioavailability of the compound within the nasal cavity, thereby optimizing its therapeutic potential.

Comments 3. Can the developed system be considered as a universal system and be used for the delivery of other drugs?

Response 3. Thank you for your question. Thermosensitive and mucoadhesive hydrogels, such as the one developed in this work, are being extensively investigated as alternative formulations for the delivery of a wide range of drugs. These include antimicrobials, analgesics, and anticancer agents, among others (3-5). Therefore, the developed system can indeed be considered a universal platform for drug delivery applications.

Comments 4. Using a developed gel with a psychedelic drug raises a number of medical questions: In the event of an overdose or unpredictable reaction by the patient, can the gel be quickly removed? How to control the gel's distribution in the sinuses? How to assess how much of the substance has entered the body and how much has leaked out of the nose to the outside?

Response 4. It is important to emphasize that this work is not aimed at the clinical stage, but rather at the characterization of a formulation that could eventually become a therapeutic alternative. During the development of a drug, one of the critical stages is determining the dosage, its efficacy, and its safety for the patient. Although adverse reactions and overdoses can occur with any medication, these are studied in the preclinical stage, including potential contingencies for the issues mentioned by the reviewer. In this particular case, since it involves a tryptamine, its use should be restricted specifically to healthcare professionals, making the risk of overdose unlikely. As for side effects, they will vary from patient to patient, and it is the treating physician’s responsibility to select the best therapy to mitigate them.

Regarding the removal of the gel, it is likely possible through suction, similar to the way mucus associated with nasal obstructions is aspirated. Overall, the nasal route is relatively easy to access, unlike, for example, the intravenous route, for which tryptamine formulations are also being developed (6).

The distribution of the hydrogel in the nasal cavity correlates with the type of spray used to apply the formulation. The angle of the spray cone is a function of the hydrogel's polymeric component ratios, and thus is related to viscosity. Depending on the angle, deposition will occur in different areas of the nasopharynx. Specifically, for hydrogels with a viscosity similar to the one we developed, the spray angle should be around 30° in order to obtain greater deposition in the nasal turbinate zone, according to scientific literature (7).

The amount of drug that enters the bloodstream can be easily determined through pharmacokinetic assays. Additionally, the quantity that reaches the Central Nervous System can be evaluated during preclinical trials in animals by quantifying the drug in brain homogenate. By considering both values, it is possible to determine the amount lost and, therefore, the bioavailability.

Comments 5. Are there other intranasal prolonged release systems in medicine?

Response 5. Apart from hydrogels, there are other systems that achieve intranasal controlled release over time. These include nanoparticles or liposomes, which encapsulate the drug to manage its release rate, ensuring steady and prolonged delivery. Mucoadhesive polymers are another example; they adhere to the nasal mucosa, providing a gradual release of the drug as they slowly break down. Microspheres are also utilized, incorporating the drug within their structure and releasing it gradually as the microspheres degrade. The use of these technologies aims to improve drug efficacy, reduce the frequency of dosing, enhance patient compliance, and minimize side effects by maintaining more consistent drug levels within the body. However, to the best of our knowledge, there are currently no sustained release intranasal formulations approved by regulatory authorities. Nevertheless, this is a particularly active research area in the field of pharmaceutical dosage forms development.

Comments 6. How are the selected polymers (P407, P188, HPMC) currently used in medicine? Are there any examples of the use of a blend of these polymers?

Response 6. According to FDA’s “Inactive Ingredient Search for Approved Drug Products”, there exist several formulations currently marketed, containing Poloxamer P407 Poloxamer P 188 and HPMC respectively. The selected polymers are widely utilized in medicine for their distinct properties and functionalities. Poloxamer 407 acts as a viscosant agent and exhibits thermosensitive properties, forming gels at body temperature, which is advantageous for sustained or controlled drug delivery systems. It also favors drug solubility and stability in formulations (8-10). Poloxamer 188 provides solubility enhancement of poorly soluble drugs and is also employed in cell cryopreservation to protect cells during freezing and thawing, with potential applications in tissue repair (9, 10). HPMC acts as a mucoadhesive polymer in nasal, ophthalmic, and oral drug delivery, prolonging drug residence at the administration site (11). Blends of P407, P188, and HPMC are commonly used to create thermosensitive and mucoadhesive hydrogels, optimizing their combined properties to transition from liquid to gel at body temperature, adhere to mucosal surfaces, and release drugs in a controlled manner. These blends illustrate the versatility and effectiveness of combining these polymers in pharmaceutical formulations for enhanced therapeutic outcomes (12).

Comments 7. Are there any studies of the effects of these polymers on 5-MeO-DMT?

Response 7. To the best of our knowledge, no studies have addressed this point. However, a long-term stability study could be conducted either on the formulation itself or through binary mixtures of the drug and excipients separately.

In reading the main body of the article, the following questions and comments arose:

Comments 8. What is "PE"? (line 10) This abbreviation occurs only once and is not deciphered

Response 8. Thank you for pointing that out. The occurrence of "PE" in line 10 was a typographical error.

Comments 9. Paragraph experiment design (3.1). The authors use mass-to-volume ratios (w/v %) as the main characteristic for polymer composition selection. This characteristic is well suited for dissolution of substances in pure water. Perhaps for a multicomponent system it is better to use mass fraction (w/w %). This approach may be important because Section 2.2 shows that the developed system contains glycerin, sodium chloride, benzalkonium chloride, 5-MeO-DMT and water. As the concentration of the main polymers increases, the mass fraction of the other components of the system changes. It turns out that the other components are variables and they are left out of the authors' view. For example, viscosity is affected by the mass fraction of water, while Tgel can be affected by the mass fraction of salt. This issue can be avoided if the total mass of P407, P188, HPMC remains constant in all experiments. The authors should give arguments why the ratio of the other components of the system is not important.

Response 9. Formulation optimization involved varying the concentrations of polymers P407, P188, and HPMC, while keeping all other components constant. This decision was based on the assumption that the variations mentioned by the reviewer would have negligible effects. The correlation results observed in the response surfaces of the experimental design supported this assumption. Specifically, the coefficients associated with the polymers exhibited highly significant p-values, affirming their critical importance in the model. Furthermore, the r-squared value approached 1, indicating that the potential impact of any other factor would be minimal. The only component besides the polymers that could potentially exert significant influence is 5-MeO-DMT itself; however, the study predetermined a fixed dose for consistency throughout. The primary objective was not to optimize the dose of 5-MeO-DMT, but rather to adjust the polymers to meet the requirements for nasal application.

Comments 10. How were the starting concentrations of polymers at center points chosen? (P407=17.5; P188=10; HPMC=0.55)

Response 10. The experimental design workspace was chosen to ensure both the technological feasibility of hydrogel manufacturing and adherence to the targeted goals for nasal application regarding viscosity, gelation temperature, and mucoadhesion values. The alignment with these goals was determined based on relevant bibliographical reports (13, 14) and previous exploratory test runs.

Comment 11. Figure 2 may become clearer if you sign which curve refers to which polymer

Response 11. Figure 2 does not depict individual factors of the experimental design separately. Rather, it displays the predicted values obtained by the model for each of the responses (viscosity, Tgel, and mucoadhesion).

Comment 12. On lines 327-328, the authors write "In Figure 6, Raman spectra of P407 (a), P188 (b), HPMC (c), 5-MeO-DMT (d), Glycerol (e), and the optimized hydrogel are presented", but in Figure 6 there are no letters a-e and no Glycerol is present

Response 12. Regarding lines 327-328, we acknowledge the discrepancy noted. The Raman spectroscopy study did not actually include glycerol, benzalkonium chloride, or sodium chloride for the same reason: these components were considered minor within the total gel mass and likely below the detection limit of our equipment. The mention of glycerol in the figure caption was an error and should not have been included. Our focus was specifically on components that could potentially influence the outcomes of the experimental design.

Comments 13. Line 340 has a reference to Figure 10, but there is no such figure present

Response 13. Thank you for bringing this to our attention. The reference to Figure 10 on line 340 was an error. We apologize for any confusion caused. There is no Figure 10 in the manuscript. The correct reference should have been to Figure 6.

Comments 14. In 3.4 the authors study the release of the substance from the gel. It is written that 73.76% was released, but it is not clear whether this is enough or not for medical use. The section will be richer if the authors write how much 5-MeO-DMT is usually injected into a patient and how much gel is needed for this amount of substance.

Response 14. The dosage applied in clinical settings will depend on the specific pathology being treated. Once determined, the hydrogel formulation can be adjusted accordingly. The primary focus of this study was to develop a thermosensitive and mucoadhesive hydrogel containing a sufficient amount of 5-MeO-DMT to act as a reservoir for controlled release. Subsequent studies will establish safety and efficacy levels to fine-tune the dosage.

References

  1. Good M, Joel Z, Benway T, Routledge C, Timmermann C, Erritzoe D, et al. Pharmacokinetics of N,N-dimethyltryptamine in Humans. Eur J Drug Metab Pharmacokinet [Internet]. 2023;48(3):311–27. Available from: https://doi.org/10.1007/s13318-023-00822-y
  2. Shen HW, Jiang XL, Yu AM. Nonlinear pharmacokinetics of 5-methoxy-N,N-dimethyltryptamine in mice. Drug Metab Dispos. 2011;39(7):1227–34.
  3. Ortega A, da Silva AB, da Costa LM, Zatta KC, Onzi GR, da Fonseca FN, et al. Thermosensitive and mucoadhesive hydrogel containing curcumin-loaded lipid-core nanocapsules coated with chitosan for the treatment of oral squamous cell carcinoma. Drug Deliv Transl Res [Internet]. 2023;13(2):642–57. Available from: https://doi.org/10.1007/s13346-022-01227-1
  4. Thouvenin A, Toussaint B, Marinovic J, Gilles AL, Dufaÿ Wojcicki A, Boudy V. Development of Thermosensitive and Mucoadhesive Hydrogel for Buccal Delivery of (S)-Ketamine. Pharmaceutics. 2022;14(10).
  5. Permana AD, Asri RM, Amir MN, Himawan A, Arjuna A, Juniarti N, et al. Development of Thermoresponsive Hydrogels with Mucoadhesion Properties Loaded with Metronidazole Gel-Flakes for Improved Bacterial Vaginosis Treatment. Pharmaceutics. 2023;15(5).
  6. Vogt SB, Ley L, Erne L, Straumann I, Becker AM, Klaiber A, et al. Acute effects of intravenous DMT in a randomized placebo-controlled study in healthy participants. Transl Psychiatry. 2023;13(1):1–9.
  7. Nižić L, Ugrina I, Špoljarić D, Saršon V, Kučuk MS, Pepić I, et al. Innovative sprayable in situ gelling fluticasone suspension: Development and optimization of nasal deposition. Int J Pharm. 2019;563(April):445–56.
  8. Kabanov, A. V., & Alakhov, V. Y. (2002). Pluronic block copolymers in drug delivery: From micellar nanocontainers to biological response modifiers. Crit. Rev. Ther. Drug Carrier Syst. 2002;19(1):1-72. DOI: 10.1615/critrevtherdrugcarriersyst.v19.i1.10
  9. H. Omidian, R. L. Wilson. Long-Acting Gel Formulations: Advancing Drug Delivery across Diverse Therapeutic Areas. Pharmaceuticals. 2024, 17(4), 493; DOI: org/10.3390/ph17040493 11.
  10. R. Fan, Y. Cheng, R. Wang, T. Zhang, H. Zhang, J. Li, S. Song, A. Zheng. Thermosensitive Hydrogels and Advances in Their Application in Disease Therapy. Polymers. 2022; 14(12): 2379. DOI: 10.3390/polym14122379
  11. B. M. Boddupalli, Z. N. K. Mohammed, R. A. Nath, D. Banji Mucoadhesive drug delivery system: An overview. J. Adv. Pharm. Technol. Res. 2010; 1(4): 381–387. DOI: 10.4103/0110-5558.76436
  12. Ruel-Gariépy, E., Leroux, J. C. In situ-forming hydrogels—review of temperature-sensitive systems. European Journal of Pharmaceutics and Biopharmaceutics, 2004, 58(2), 409-426. DOI: 10.1016/j.ejpb.2004.03.019
  13. ONGUN M, TUNÇEL E, KODAN E, TUĞCU DEMİRÖZ FN, TIRNAKSIZ FF. Development and Characterization of Mucoadhesive-Thermosensitive Buccal Gel Containing Metronidazole for the Treatment of Oral Mucositis. Ankara Univ Eczac Fak Derg. 2020;44(3):517–39.
  14. M.A. Fathalla Z, Vangala A, Longman M, Khaled KA, Hussein AK, El-Garhy OH, et al. Poloxamer-based thermoresponsive ketorolac tromethamine in situ gel preparations: Design, characterisation, toxicity and transcorneal permeation studies. Eur J Pharm Biopharm. 2017;114:119–34.

Reviewer 2 Report

Comments and Suggestions for Authors

The manuscripts study novel thermosensitive and mucoadhesive nasal hydrogel containing 5-MeO-DMT.

General

The manuscript must be formatted.

“Box-Behnken” and “Optimization” must be added to the title.

How do you measure the preservation of ciliary functioning?

Information about the contact time between the hydrogel and the mucosa must be measured and discussed in detail.

Abstract

Please add future use and expectations of the product produced.

Keywords

Do not repeat similar keywords to the title. Also, remove misspelled keywords.

Introduction

Split the introduction to several paragraphs. Each paragraph must contain one problem statement, and the last paragraph must emphasise the objective and hypothesis of the study.

Define “thermosensitive” in the introduction part. Elaborate and give examples of hydrogel sensitivity towards thermal and how they react physically, chemically, and mechanically.

Please add examples of different hydrogel sensitivity based on gelation temperature, viscosity and mucoadhesion. Please relate them based on a strong gathering of information from past publications.

Please add an introduction of 5-MeO-DMT based on their sensitivity to the thermal and relation to the hydrogel.

Materials and Methods

Please use the correct symbol for each unit.

Please add a number of repetitions to the related analysis.

Conclusion

Lengthy and not answering the objective of the study.

Comments on the Quality of English Language

English writing can be improved in terms of technicality and cohesiveness.

Author Response

Reviewer #2

Comments 1. “Box-Behnken” and “Optimization” must be added to the title.

Response 1. “Box-Behnken” and “Optimization” were added to the title.

Comments 2. How do you measure the preservation of ciliary functioning?

Response 2. We appreciate the reviewer's suggestions. We have revised both the introduction and the title accordingly. As for evaluating the ciliary function, it is currently beyond the scope of this study but may be considered in future stages. In particular, measurements require state of the art techniques such as ciliary beat pattern and ciliary beat frequency, which have to be performed by highly specialized personnel.

Comments 3. Information about the contact time between the hydrogel and the mucosa must be measured and discussed in detail.

Response 3. We agree with the reviewer about studying contact time, since it is of paramount importance. However, again, this is completely out of the scope of our work, since we intended to first develop a candidate formulation for future testing. In order to accurately measure contact time between the hydrogel and the mucosa, a few methods could be tried out, all of them involving somehow or another imaging techniques associated with radio or fluorescent labeling. For instance, a chamber mimicking nasal mucosa with a controlled flow of simulated nasal fluid could be set up, and the time the hydrogel remains attached before being washed away by the fluid measured. Another way could be performing a similar experiment using ex vivo tissue submerged in simulated nasal fluid. Both approaches rely on measuring the amount of labeling agent remaining after the wash out. However, it also assumes that such an agent would not be released from the matrix by diffusion, just by the action of erosion both intrinsic or extrinsic to the matrix. Marking the hydrogel could also potentially have an impact on the formulation’s own performance.  In light of all of the above, in vivo methods seem to be the most accurate option, because it also involves ciliary clearance (highlighted above by the reviewer). A possible approach would be to administer the gel intranasally to lab animals and then monitor the amount present by imaging techniques such as PET-CT or even performing histological analysis.

Comments 4. Please add future use and expectations of the product produced.

Response 4. Thank you for your feedback. We have incorporated the future use and expectations of the product in our revision.

Comments 5. Do not repeat similar keywords to the title. Also, remove misspelled keywords.

Response 5. As suggested, we have changed the keywords as requested.

Comment 5. Split the introduction to several paragraphs. Each paragraph must contain one problem statement, and the last paragraph must emphasize the objective and hypothesis of the study.

Define “thermosensitive” in the introduction part. Elaborate and give examples of hydrogel sensitivity towards thermal and how they react physically, chemically, and mechanically.

Please add examples of different hydrogel sensitivity based on gelation temperature, viscosity and mucoadhesion. Please relate them based on a strong gathering of information from past publications.

Please add an introduction of 5-MeO-DMT based on their sensitivity to the thermal and relation to the hydrogel.

Response 5. Thank you for the suggestion. The introduction has been updated in the revised version of the manuscript.

Considering that 5-MeO-DMT has a melting point of 66 °C, it should not be sensitive to the temperature changes our hydrogel formulations were subjected to. Moreover, please bear in mind that these hydrogels are intended to use at body temperatures, and therefore their purpose is not to protect the drug against temperature changes.

Comments 6. Please use the correct symbol for each unit.

Response 6. We have changed grams to g, hours to h and seconds to s.

Comments 7. Please add a number of repetitions to the related analysis.

Response 7. Thanks for pointing this out. The number of repetitions were added to the in vitro release study, in accordance with the results section.

Comments 8. Lengthy and not answering the objective of the study.

Response 8. We have changed the conclusion accordingly.

Round 2

Reviewer 2 Report

Comments and Suggestions for Authors

The article has been improved accordingly.